# A Computer-Aided Diagnostic System to Identify Diabetic Retinopathy, Utilizing a Modified Compact Convolutional Transformer and Low-Resolution Images to Reduce Computation Time

**DOI:** 10.3390/biomedicines11061566

**Published:** 2023-05-28

**Authors:** Inam Ullah Khan, Mohaimenul Azam Khan Raiaan, Kaniz Fatema, Sami Azam, Rafi ur Rashid, Saddam Hossain Mukta, Mirjam Jonkman, Friso De Boer

**Affiliations:** 1Health Informatics Research Lab, Department of Computer Science and Engineering, Daffodil International University, Dhaka 1207, Bangladesh; inam15-2575@diu.edu.bd (I.U.K.); kaniz15-12344@diu.edu.bd (K.F.); 2Department of Computer Science and Engineering, United International University, Dhaka 1212, Bangladesh; mraiaan191228@bscse.uiu.ac.bd (M.A.K.R.); saddam@cse.uiu.ac.bd (S.H.M.); 3Faculty of Science and Technology, Charles Darwin University, Darwin, NT 0909, Australia; mirjam.jonkman@cdu.edu.au (M.J.); friso.deboer@cdu.edu.au (F.D.B.); 4Department of Computer Science and Engineering, Penn State University, State College, PA 16801, USA; rafiurrashid150@gmail.com

**Keywords:** diabetic retinopathy, retinal fundus images, image pre-processing, compact convolutional transformer, ablation study, low pixel

## Abstract

Diabetic retinopathy (DR) is the foremost cause of blindness in people with diabetes worldwide, and early diagnosis is essential for effective treatment. Unfortunately, the present DR screening method requires the skill of ophthalmologists and is time-consuming. In this study, we present an automated system for DR severity classification employing the fine-tuned Compact Convolutional Transformer (CCT) model to overcome these issues. We assembled five datasets to generate a more extensive dataset containing 53,185 raw images. Various image pre-processing techniques and 12 types of augmentation procedures were applied to improve image quality and create a massive dataset. A new DR-CCTNet model is proposed. It is a modification of the original CCT model to address training time concerns and work with a large amount of data. Our proposed model delivers excellent accuracy even with low-pixel images and still has strong performance with fewer images, indicating that the model is robust. We compare our model’s performance with transfer learning models such as VGG19, VGG16, MobileNetV2, and ResNet50. The test accuracy of the VGG19, ResNet50, VGG16, and MobileNetV2 were, respectively, 72.88%, 76.67%, 73.22%, and 71.98%. Our proposed DR-CCTNet model to classify DR outperformed all of these with a 90.17% test accuracy. This approach provides a novel and efficient method for the detection of DR, which may lower the burden on ophthalmologists and expedite treatment for patients.

## 1. Introduction

Diabetic retinopathy (DR) is a progressive condition, often affecting both eyes. It is a complication of diabetes mellitus, which occurs when high glucose levels in the blood induce lesions on the retina of the eye. It is the leading cause of blindness among people with diabetes [1]. According to the World Health Organization (WHO), in 2014, 422 million people worldwide had diabetes; a total of 35% of these had developed retinopathy due to the progressive destruction of blood vessels in the retina [2]. Irreversible damage to the blood vessels can occur if this persists for a prolonged period [3]. Early diagnosis of diabetic retinopathy can inform treatment and prevent DR from reaching the most severe stage. However, based on years of experience and time, ophthalmologists’ clinical expertise is required for effective diabetic retinopathy screening [4]. DR can be diagnosed and classified employing color fundus images. The DR screening process involves an ophthalmologist examining the fundus and integrated retinal images captured by specialized equipment, which requires significant time. There is a severe shortage of ophthalmologists, so there are fewer available screening appointments than individuals in need [3]. Automatic diabetic retinopathy classification techniques may help diagnose DR and increase screening productivity [5].

Computer aided diagnosis (CAD) systems for diverse medical imaging modalities are currently being studied [6]. Researchers have been introducing many aspects to diagnosing diabetic-related diseases in recent times [7]. However, deep learning models have several shortcomings, including complexity and a long training time. While convolution is often used in classification models, it is possible to construct effective models without this. Studies of transformer models have gained prominence in the field of Machine Learning (ML). Adding simple convolutional blocks to the tokenization stage of a visual transformer was the idea behind the Compact Convolutional Transformer (CCT), which was first proposed by Hassani et al. [8]. To address the shortcomings (i.e., high training time, complexity, high dimensional image) of the deep learning models, this study proposes a novel automated approach to classify the DR with small image size and short training time to detect the progression of DR at an early stage.

Several researchers have classified DR using different retinal fundus datasets, deep learning (DL), and ML techniques. In most instances, their methods failed to provide a strategy for dealing with a large data hub containing low-pixel images. We describe the limitations of these models in the literature review. We attempt to overcome these limitations by employing the CCT model on a merged dataset comprising many diabetic retinopathy photos and classify DR into five classes: no DR (Grade 0), mild Non-proliferative Diabetic Retinopathy (NPDR) (Grade 1), moderate NPDR (Grade 2), severe NPDR (Grade 3), and Proliferative Diabetic Retinopathy (PDR) (Grade 4).

The main contributions of this paper are as follows:A total of five datasets are amalgamated to generate a larger and more encompassing dataset, and a wide range of diabetic retinopathy images with diverse resolutions and image quality are utilized. The resulting datahub contains 53,185 raw images.Several image pre-processing techniques, including Otsu thresholding, contour detection, region of interest (ROI) extraction, morphological opening, non-local means denoising, and Contrast Limited Adaptive Histogram Equalization (CLAHE), are used to eliminate artifacts and noise from retinal fundus images and enhance their quality.An augmentation strategy is applied to increase the number of images and create a large data hub.A detailed comparison between three transformer models (Vision transformer, Swin transformer, and Compact Convolutional Transformer) and four transfer learning models (VGG19, VGG16, MobileNetV2, and ResNet50) using our dataset is done to evaluate how well the models perform in terms of accuracy and training time and to find the optimal transformer or transfer learning model for a vast number of images.A new model, DR-CCTNet, is proposed. It is constructed by modifying the original CCT model to overcome the long training times issue and work with a large data hub. Using convolutional blocks, the tokenization step of the vision transformer is accomplished, drastically lowering model training time while attaining high accuracy, even with low-pixel images.An ablation study is carried out by modifying the proposed model’s various hyper-parameters and layer architecture to improve its performance further, reducing the number of parameters and making the training process less complicated and time-consuming.To further examine the generalization capabilities and robustness of our model in relation to the size of the training dataset, the model is trained four times with a gradually decreasing number of images. Even with a smaller number of images, the model demonstrates good performance, demonstrating the robustness of the DR-CCTNet model.

## 2. Literature Review

In recent years, the need for accurate diagnosis of DR has attracted attention. Several CAD methods have been developed to aid clinicians’ analysis of fundus images [9]. Deep learning algorithms are becoming more popular due to their superiority in automatic feature extraction and classification. However, deep learning has drawbacks when working with big data hubs because of its computational complexity [6]. Machine Learning models, transfer learning models, and in some cases, transformers have been described for DR classification. In this section, prior studies related to DR classification are discussed. Hemanth et al. [10] proposed effective image-processing techniques with histogram equalization and contrast-enhanced limited adaptive histogram equalization techniques. They employed a CNN model for classification and used 400 retinal fundus images of the MESSIDOR dataset. Their model achieved 97% accuracy, 94% recall, 98% specificity, 94% precision, an F1-score of 94%, and a mean value of 95%. Other researchers [3] focused on categorizing DR fundus images according to the severity of the diseases, achieving high accuracy. They used the Alexnet model architecture to deploy suitable pooling, Softmax, and rectified linear activation unit (ReLU) functions. They utilized 1190 retinal images of the Messidor dataset, including healthy retinas, DR stage 1, DR stage 2, and DR stage 3. They split their images into RGB channels and used the green channel for further processing. Classification accuracies of 96.6%, 96.6%, 95.6%, and 96.6% were achieved for healthy images, stage 1, stage 2, and stage 3, respectively. Gu et al. [11] proposed a method to detect all five DR stages, no DR, mild, moderate, severe, and proliferative DR. They employed a feature extraction block (FEB) and a grading prediction block (GPB) and used the DDR and the IDRiD datasets. The area under the curve (AUC) for classes 0 through 5 were 0.9980, 0.6128, 0.9509, 0.9455, 0.9741, and 0.9293, respectively. Sehrish et al. [12] proposed a CNN ensemble-based framework for identifying and classifying the various stages of DR in color fundus images. They applied up-and-down sampling to the Kaggle DR dataset to balance the dataset for multiclass classification. They integrated multiple deep learning models, including Reset50, inceptionV3, Xception, Dense121, and Dense 169, and trained these on balanced and unbalanced datasets. A test dataset was developed to validate the model, and the findings were compared to previous research using a comparable dataset. However, the accuracy of their proposed ensemble model for the unbalanced dataset was just 80.8%.

Liu et al. [13] introduced three hybrid deep learning models, Hybrid-a, Hybrid-f, and Hybrid-c, for diabetic retinopathy classification with an improved loss function. To assess the proposed model’s performance, they utilized three datasets, DeepDR, APOTS, and EyePACS. Binary processing, resizing, and geometric augmentation techniques were used to process the dataset. The improved version of the E-CE loss function accelerated their training process. Their proposed model structures outperformed multiple base models, attaining an accuracy of 86.4%. Wu et al. [14] created a CNN-based Coarse-to-Fine Network (CF-DRNet) architecture for automatic DR categorization. They utilized two publicly available datasets, IDRiD and the Kaggle fundus image dataset, and employed augmentation strategies to overcome the class imbalance problem. ResNet, a transfer learning model, was employed for comparison. Their model outperformed ResNet with an accuracy of 80.61% and 80.00% for the IDRiD and the Kaggle fundus image dataset, respectively. Lam et al. [15] applied convolutional neural networks (CNNs) to color fundus images of the Kaggle and Messidor datasets for automated diabetic retinopathy detection. The CLAHE algorithm improves the dataset quality, and various augmentation techniques help balance the dataset. The GoogleNet transfer learning model provided the best accuracy of 74.5% in binary classification, and the proposed model obtained accuracies of 68.8% and 57.2% on three and four-class classifications, respectively. Gao et al. [16] proposed a hybrid structure of different inception models. They used three normalization techniques and five augmentation strategies to pre-process the fundus images. Their proposed model inception@4 attained an accuracy of 88.72% and a recall value of 94.84% for multiclass classification.

The existing literature shows that most researchers have worked with a limited number of images, and a minimum image size is 150 × 255 pixels, as can be seen in Table 1. To address these limitations, we introduce an efficient approach to work with a data hub of 154,882 images. Low resolution images with dimensions of 16 ×16 pixels are used.

## 3. Methodology

This research consists of six stages. Figure 1 illustrates the overall procedure where the first five public datasets have been merged to create a large dataset. In the second stage, the fundus photos are pre-processed to eradicate noise and artifacts and enhance the quality of the images. Under-sampling and augmentation techniques are utilized in the third stage to create a data hub. Stage four of this study involves constructing a base model, and in stage five, ablation studies for this model are done. Finally, we evaluate the performance of the model and other results in stage six. All the stages are described in detail in this section.

### 3.1. Dataset Description

One of the major objectives of our study is to work with a wide range of images with different resolutions and quality to make a diverse dataset. We have integrated five datasets: APTOS [17], Messidor2 [18], IDRiD [19], DDR [20], and Kaggle Diabetic Retinopathy [21]. Every dataset has five categories: no DR (Grade 0), mild NPDR (Grade 1), moderate NPDR (Grade 2), severe NPDR (Grade 3), and PDR (Grade 4). Grade 0 to Grade 4 represent the progression of diabetic retinopathy. Table 2 illustrates the number of images in each dataset and the number of images for each grade after merging all the datasets. Some of the fundus images of each dataset are displayed in Figure 2.

### 3.2. Image Pre-Processing

Image pre-processing is essential to improve the quality of the images collected from public datasets. High-quality color retinal images are required for reliable and prompt classification of DR [22]. In this study, five different image datasets are combined. Public datasets of retinal fundus images have been developed with varying resolutions and compression formats and contain background noise [23]. To ensure the quality of the fundus images, we employ various methods to eliminate artifacts, eliminate undesirable noise, and enhance the image. Figure 3 illustrates the process of image pre-processing.

#### 3.2.1. Artifacts Removal

Images can contain artifacts, such as unwanted regions or objects. Removing these artifacts, especially if they are evident, is crucial for DR classification. Our dataset of retinal fundus images contains a considerable amount of black background, which is unnecessary for the classification task. We eliminate the black region from our dataset using three procedures (e.g., Otsu thresholding, contour detection, and ROI extraction).

##### Otsu Thresholding

The Otsu method determines an optimal global threshold value using an image’s histogram [24]. The threshold is used to distinguish the image’s background and the ROI. This nonlinear transformation converts a grayscale image to a binary image. For our study, we use a minimum value of 0 and a maximum value of 255. Utilizing the Otsu algorithm, we successfully distinguish the background and the ROI. Figure 3B(i) exhibits the Otsu threshold mask.

##### Contour Detection

A contour is an outline that depicts the shape or form of an object, and contour detection is a technique that extracts curves from images that correspond to the shape of the objects [25]. The binarized image from Otsu’s thresholding is used as the source image for the contour detection algorithm. Figure 3B(ii) represents the contour of the fundus image, which is necessary for the next step.

##### Regions of Interest Extraction

In the image of the retinal fundus, the ROI is the targeting area for classifying diabetic retinopathy. The ROI is cropped using pixel coordinates and the contours list from the previous stage. Figure 3B(iii) depicts the separation of the ROI region without unnecessary black background padding.

#### 3.2.2. Noise Eradication

The publicly available fundus images dataset contains noise, making recognizing and evaluating retinal fundus images challenging [23]. To eliminate noise from the images in the dataset, we first employ morphological opening, and then non-local means denoising. Figure 3C provides a pictorial representation of the noise elimination steps.

##### Morphological Opening

Morphological opening is utilized to eliminate image artifacts. Morphological opening is a method that removes single-pixel noise, such as noisy spikes and tiny spurs, and blackens small objects [6]. Objects usually retain their original dimensions and shapes. For the input image, the initial step in this operation is erosion, followed by dilation. After getting the ROI extracted image, the morphological opening is applied using a kernel. Several kernel sizes were tested, and a kernel size of (10,10) resulted in the best results. A resultant image after the morphological opening procedure is provided in Figure 3C(i).

##### Non-Local Means Denoising (NLMD)

NLMD has been adopted to reduce underlying noises in our study. The NLMD approach replaces each pixel with the weighted average of nearby pixels. The formula for this algorithm is as follows [26]:(1)$$NLu\left(p\right)=\frac{1}{C\left(p\right)}\int f(d\left(B\left(p\right),B\left(q\right)\right)u\left(q\right)dq$$

Here, NLu(p) is the filtered value, f  is a decreasing function, d(B(p), B(q))  is the Euclidean distance between patches (p) and (q) which specifies the pixel’s surroundings, and C(p) is a normalizing factor that divides the average weighted function result to obtain the filtered value. In our study, we have performed NLMD using the fastNlMeansDenoisingColored() function. The output image after morphological opening is utilized as the input image for NLMD. The templateWindowSize and searchWindowSize are set at 7 and 21, respectively. Our approach eliminates the underlying noise in the retinal fundus images in our study, resulting in an adequately denoised image. Figure 3C(ii) depicts a noise-free image produced using NLMD.

#### 3.2.3. Image Enhancement (CLAHE)

Denoised retinal fundus images are processed with CLAHE [27] to enhance the contrast. Darkness, brightness, and uneven illumination distort the image [28]. To address this, we converted the RGB color channels to YUV color channels, where the Y channel (Figure 3D) represents the lightness components [29]. Several different parameter combinations were utilized to tune clipLimit and tileGridSize, and it was found that a clipLimit of 0.5 and a tileGridSize of (8,8) give the best results. These values are utilized when applying CLAHE to the Y channel. The final image after CLAHE processing is shown in Figure 3D.

### 3.3. Data Balancing

After completing the image preprocessing steps, a vital concern is creating a large datahub without making the dataset biased towards any particular class and creating a significant data imbalance between classes. To achieve this, the random under sampling technique is applied to the majority classes to balance the dataset. Subsequently, 12 augmentation strategies are utilized to increase the size of the dataset and create a large data hub.

#### 3.3.1. Under Sampling

Table 2 shows large differences in the number of images in different classes. Grade 0 and Grade 2 have more images than the other classes. To resolve these discrepancies, we utilize a unique strategy to reduce the number of images in the two classes with the most significant number of images. Algorithm 1 describes the process in detail.
**Algorithm 1: Under Sampling**1. Define class names*class_names = [‘Grade 0’, ‘Grade 1’, ‘Grade 2, ‘Grade 3’, ‘Grade 4’]*2. Define image numbers for each class*class_image_numbers = [7 ‘Grade 3’: 1441, ‘Grade 4’: 1987]*3. Calculate the average image numbers of the smaller three classes*smallest_classes = [‘Grade 1’, ‘Grade 3’, ‘Grade 4’]**Smaller classes image numbers = [class_image_numbers[class_names] for class_name in smaller_classes]**average_image_numbers = int(np.mean(lower_classes_image_numbers))*4. Random under sample the higher two classes to the average size*Larger_classes = [‘Grade 0’, ‘Grade 2’]**for class_name in larger_classes**class_image_number = class_image_numbers[class_names]**undersample_ratio = average_image_numbers/class_image_number**undersampled_image_numbers = int(class_image_number* × *undersample_ratio)**class_image_numbers[class_name] = undersampled_image_numbers*

Algorithm 1 describes that we calculated the mean number of images in the three smaller classes, which is 2384. After getting the mean, we set this value as the threshold for random under sampling and reduced the number of images to 2384. Table 3 depicts the number of images after under sampling.

#### 3.3.2. Data Augmentation

Data augmentation can help to balance a dataset. After reducing the two majority class images, we employed a data augmentation method to create a balanced dataset in this study. Data augmentation is also a fundamental practice for avoiding overfitting problems [30]. Increasing the amount of data should create samples similar to actual images and ensure the dataset is balanced. Another issue is maintaining high image quality, which is especially important in the medical field [31]. In this work, we have employed 12 types of augmentation techniques to enhance the dataset. These techniques included two types of zooming (with factors of 1.2 and 1.5), two types of flipping (vertical and horizontal plus vertical), rotations of 90 degrees and −90 degrees, and adjustments to the brightness, sharpness, contrast, and color, as well as color jittering and gamma correction. We used these augmentation techniques to generate a large data hub of quality fundus images.

Changing the image’s brightness, sharpness, contrast, and color can improve image quality and increase the visibility of structures in fundus images. Figure 4B–E depicts fundus images acquired through these methods. Adjusting the image properties improves the visibility of substantial structures in fundus images, including blood vessels, optic discs, and lesions.

In this study, gamma correction is employed to adjust an image’s gamma value, which affects the image’s brightness and contrast. This technique also improves the visibility of structures in images, as depicted in Figure 4F. Gamma correction can enhance the contrast of fundus images, making the forms more distinguishable and facilitating accurate feature extraction.

Color jittering is a method used to add random color variations to an image. We applied color jittering to fundus images to diversify the dataset’s color palette. This method could enhance the model’s ability to recognize structures in fundus images with varying hues. Color jittering can simulate the natural color variation of fundus images caused by variations in illumination, camera settings, and patient characteristics, thereby making the model more robust to these differences. The image presented in Figure 4G is the result of color jittering.

Zooming is a technique used to increase or decrease the scale of an image. Zooming in enlarges the fundus image, enabling a closer examination of details, whereas zooming out reduces the image size, providing a broader view of the retina. It may enhance the visibility of delicate structures in fundus images, such as blood vessels, optic discs, and lesions, by providing various scales of the same image. Fundus images are zoomed with factors of 1.2 and 1.5 in this study. Figure 4H,I portray images zoomed with a factor of 1.5 and 1.2, respectively.

Rotation is a technique where a certain angle rotates an image. For our dataset, we performed two forms of rotation: a 90-degree and a −90-degree rotation. These rotations provide various orientations of the same fundus image, allowing the model to learn features from different angles. This can improve the model’s ability to recognize structures in fundus images with varying orientations. Figure 4J,K depict images rotated by 90 and −90 degrees, respectively.

An approach that horizontally or vertically mirrors an image is called “flipping.” Fundus images have been flipped in two ways, vertically and horizontally, in our study. Flipping fundus images horizontally or vertically may enhance the model’s ability to generalize various image orientations encountered during testing. Figure 4L depicts vertical flipping, whereas Figure 4M depicts vertical plus horizontal flipping.

Table 3 lists the number of images of the dataset after completing 12 augmentation strategies.

### 3.4. Model Comparison

#### 3.4.1. Training Strategy for Transfer Learning Models

To train the models, the batch size was set to 128, and the number of epochs was set to 400. The SGD optimizer and the average pooling layer were utilized with a learning rate of 0.006. In multi-class scenarios, the default loss function is mean squared error (MSE). The tanh activation was used to estimate the probability for each class. The dataset split ratio for training, validation, and testing was 70%, 20%, and 10%, respectively.

#### 3.4.2. Transfer Learning Models

A total of four distinct transfer learning models, VGG19, VGG16, MobileNetV2, and ResNet50, were used. Below is a description of these models.

##### VGG16

VGG16 is a transfer learning model consisting of 16 weighted layers, which achieved an accuracy of 92.7% on the ImageNet dataset and was part of the top five test results. The VGG can aid the kernel in learning more complicated characteristics [6].

##### VGG19

The VGG19 model is a variant of the VGG model that consists of 19 weighted layers. In addition to the VGG16 model, there are three extra FC layers with 4096, 4096, and 1000 neurons. A Softmax classification layer and five max pool layers are also present. In the convolutional layers, the ReLU activation function is employed [6].

##### ResNet50

The ResNet50 model contains 50 layers, including 48 convolutional layers and one average and max pool layer. It has about 23 million parameters [32]. The model is commonly utilized in computer vision applications because the design reduces vanishing gradient difficulties by following an alternative shortcut path. The model uses stacked convolutions to carry out convolution and max pooling layer operations. This can resolve gradient vanishing issues more effectively [33].

##### MobileNetV2

The MobileNetV2 model consists of 53 layers and has 3.5 million trainable parameters [34]. It is made up of two types of blocks, each with three layers. In both blocks, the first and the third layers are 1 × 1 convolutional layers with 32 filters, while the middle layer is a depth-based convolutional layer. The longitudinal bottlenecks between the layers are essential in preventing nonlinearity due to large data loss [27].

#### 3.4.3. Transformer Models

Nowadays, the most commonly used model for natural language processing (NLP) is the transformer model [35]. Transformer models excel in the use of attention-to-model range relationships in the data. They were designed for sequence modeling and transduction activities. The success of the transformer model in NLP has inspired researchers to extend it to computer vision, where it has shown promising results in image classification [34]. Several transformer models are also utilized in this study. The augmented dataset was employed for each of them used with three transformer models: vision transformer (ViT), shifted window transformer (Swin transformer), and compact convolutional transformer (CCT). The main objective of applying these models was to find the transformer model which would be the best base model for this study using the augmented dataset. This base model was utilized for further processing. SGD was used as an optimizer, MSE was the loss function, the learning rate was 0.06, an average pooling layer was used, the batch size was 128, and the total number of epochs was 400.

##### Vision Transformer (ViT)

Transformer models, which are widely used in natural language processing, pay attention to image patches and sequences. In this instance, a series of image patches serves as the input to the transformer block, which employs the multi-head attention layer as a self-attention mechanism. Generally, the transformer blocks provide a tensor of batch size, several patches, and projection dimensions, which can be supplied to the classifier head using Softmax to compute the class probabilities [36]. Alexey et al. [37] proposed a novel method for applying transformers to visual data and presented vision transformers (ViTs) for image categorization. Figure 5 depicts a general approach used in ViT models. It shows how an image is transformed into a group of patches, each indicating a location of a portion in the image. This approach allows us to consider an image as sequential data and use transformers originally designed for NLP. To ensure consistency in protection dimensions, these image patches are flattened before they are fed into a trainable linear projection layer. Since the ViT model is utterly indifferent to the hierarchy of the input image, position embedding is included in these projections. The transformer encoder block takes these patches, their positions, and an additional classification token, called the CLS token. To learn different types of self-attention, the transformer encoder has layers that may split their focus among seven different heads. A multi-layer perceptron (MLP) receives a combined set of results from all existing nodes.

##### Shifted Window Transformers (Swin Transformers)

Another sophisticated computer vision tool is the Swin transformer, which is a modified version of the ViT. It uses hierarchical feature maps and scales computationally with image size. It generally builds a hierarchical representation by starting with small-sized patches and gradually integrating nearby patches in deeper transformer layers. It can use feature pyramid networks (FPN) [38] or U-Net [39] for dense prediction with these hierarchical feature maps. Since the number of patches is fixed in this method, the patch size and the complexity scale linearly with the image size. Unlike previous transformer-based systems [37], which produce a single-resolution feature map and have quadratic complexity, the Swin transformer has properties that make it suitable as a general-purpose backbone for many vision tasks. Figure 6 shows the architecture of the Swin transformer. It can be seen from Figure 6 that the Swin transformer consists of four different building blocks. First, a patch partition layer will segment the input image into several patches. The patches are then sent to the Swin transformer block, which is a part of the linear embedding layer. The basic architecture is structured into four stages: linear embedding layers and transformer blocks. The Swin transformer is built based on a modified self-attention and a block that comprises multi-head self-attention (MSA), layer normalization (LN), and a 2-layer multi-layer perception (MLP) [36]. We employed the Swin transformer in this study to solve the classification problem and diagnose retinal diseases.

##### Compact convolutional transformer (CCT)

The compact convolutional transformer (CCT) has two blocks, including a transformer with sequential pooling and convolutional tokenization. The intricate workings of the CCT are depicted in Figure 7. In this process, the convolutional tokenization blocks gather image patches. The dimensions of the augmented images are the height (h), weight (w), and channel (c) of an image. The image patches are combined into a sequence of length (l). For an image (y) with dimensions h, w, and c, the convolutional tokenization procedures will be:(2)$$y_{0}=\mathrm{MaxPool}(\mathrm{ReLU}(\mathrm{Conv2D}(y)))$$

Here, the convolutional layer (Conv2D) contains 64 filters with a stride of 2 and the ReLU activation function. The maxpool layer then scales down the generated Conv2D feature maps. The convolutional tokenization block accepts images of any dimension as input. Consequently, the CCT models do not need all the image patches to be of the same size. These convolutional patches in the CNN layers aid in the model’s ability to store local spatial information. The image patches made by the first block are sent to the transformer-based backbone, where an encoder block is made up of a multi-head self-attention (MSA) layer and a multilayer perception (MLP) head. The transformer encoder uses layer normalization (LN), the GELU activation function, and dropout. In CCT models, layer normalization comes after positional embedding, where the positional embedding is learnable.

The output of the transformer backbone is pooled via the sequence pooling layer, which is utilized as an alternative to applying a class to map successive outputs to a single class [8]. This sequence pooling allows the network to weigh the sequential embedding of latent spaces generated by the transformer encoder and improve data correlation for the input data. The sequence pooling layer pools the full sequence of data since it contains meaningful information from diverse portions of the input images. This method is known as mapping transformation; it is denoted as T: ℝ(i×n×j)→ℝ(i×j).

This procedure can be described as:(3)yL=f(y0)∈ℝ(i×n×j)
where the transformer encoder of a layer is denoted as L and its output is denoted as yL or f(y0). Furthermore, a mini-batch size denoted by i, j is taken as the embedding dimension, and n indicates the sequence length. Then, yL is fed to a linear layer g (yL)∈ℝ(i×1)  and the Softmax activation function (Equation (4)) is utilized.
(4)yL′=softmax(g(yL)T)∈ℝ(i×1×n)

The output can be calculated as:(5)output;o=yL′yL=softmax(g(yL)T)×xL∈ℝ(i×1×j)

After pooling of the second dimension, output (o)∈ℝ(i×i)  is attained as an output. After passing through a linear classification layer, the images are categorized.

#### 3.4.4. Results of the Transfer Learning and Transformer Models

In this work, we initially tested four transfer learning and three transformer learning models. We evaluated our enriched dataset with an image size of 32 × 32 for each model. The number of epochs remains the same, at 400 epochs, for all transformer models. The objective of this experiment was to find the optimal model, which means that it should outperform the other models and take a minimal amount of time. Table 4 shows that the CCT model outperformed the other models for 32 × 32 images. The CCT model achieved the highest accuracy of 84.52%, taking only 78 s to complete each epoch. The ViT model had the second-highest accuracy of 81.56%, with 180 s per epoch. The Swin transformer model’s accuracy was very close to the ViT model, but it took a long time. In contrast, all the transfer learning models had low accuracies and required a long time. We can see that ResNet50 took 423 s per epoch. The remaining transfer learning models also required a long time compared to the transformer models. Since the CCT model took the least amount of time with 32 × 32 images and had the highest accuracy, we selected the CCT as the base model for the rest of this research.

### 3.5. Base Model

After testing three different transformer models, it was found that the CCT model performs very well compared to other models. There are several opportunities to enhance the CCT model’s performance. We modified the base framework of the CCT model and proposed an enhanced model. However, Figure 8 illustrates the CCT base model architecture.

The base CCT model consists of several modules and layers: an input layer, an augmentation layer, a CCT tokenizer, a multi-head attention layer, a regularization layer, and pooling, dropout, dense, and output dense layers equipped with the Softmax activation function. This framework takes input images of 32 × 32 × 3 dimensions, and several geometrical augmentations techniques are applied to the input images. The augmented images are fed to the CCT tokenizer, and all the output images are reshaped into dimensions of 36 × 128. Initially, the convolutional layer of the CCT tokenizer block has a stride size of 3, a kernel size of 5, and a kernel size of the pooling layer of 4. After passing through the CCT tokenizer, it moves to the tensor flow addons and then passes through transformer encoder block1 and block2. These encoder blocks consist of several layers, including layer normalization, regularization, multi-head attention, another pair of regularization and normalization layers, and two pairs of dense and dropout layers along with another regularization layer. The last regularization layer’s output is the encoder block’s input. The dimensions of the output of the final regularization block were 36 × 128. After the transformer encoder block comes the following layer normalization with dimensions of 36 × 128. Then the normalization output is connected to the dense layer with a Softmax layer that produces an output with dimensions of 64 × 1. This is sent to a sequence pooling layer, which generates output data of 1 × 128 dimensions. In the final steps, the retinal fundus images are classified into five classes utilizing a linear classification layer.

### 3.6. Ablation Study

As previously mentioned, we performed an ablation study on the base CTT model to optimize performance by modifying the layer design and fine-tuning the hyperparameters. A total of 12 studies were carried out, including modifying the activation functions, different pooling layers, changing the number of transformer encoder blocks, changing the kernel size, changing the size of stride, different loss functions, batch learning rates, different optimizers, and kernel size of the pooling layer. Ablation study is a sequential process. Initially, a conventional hyperparameter value is set, and then experiments are carried out by altering values. After optimizing the first parameter, a second ablation experiment is done with the optimized first parameter. This process is repeated in successive steps until we find the optimal configuration for our proposed CCT model. After performing the ablation experiments on the base CCT model, the modified CCT model has a more robust architecture with improved classification accuracy and a lower processing time. All results are listed in Table 5 and Table 6.

Study 1: Altering the image size

In this case, we tested our base model by reducing the image size. Initially, the image size was 32 × 32. After that, it was reduced to 28 × 28, 24 × 24, and 16 ×16. The main goal was to achieve high accuracy with minimal time. We can see that the CCT model achieved 83.91% accuracy using a 28 × 28 image size in 60 s. When the image size is reduced to 24 × 24, 83.23% accuracy is achieved in 48 s. When the model was tested with an image size of 16 × 16, it achieved 78.52% accuracy with a minimum training time of 80 s per epoch. The accuracies were close to the base model, but the training time was different. Although the accuracy with 16 × 16 images was 82.52%, slightly lower than the base model, and it took the minimum time per each epoch, we chose the 16 × 16 image size for further research.

Study 2: Altering the transformer layers

To attain the best accuracy, the transformer layers configuration of the base model was altered by adding or removing the transformer encoded blocks. Table 5 displays the outcomes of the model with a variety of configurations of the transformer-encoded blocks. The model attained the greatest accuracy for configuration number 1 with a training time of 70 s per epoch. However, configuration numbers 2 and 3 attained the test accuracies of 82.52% and 82.38%, respectively, which are very close to the first configuration’s acquired accuracy. Since configuration 3 had the lowest number of trainable parameters (300,678), it required the lowest number of epochs to train. Configuration 3 was therefore chosen for further ablation research.

Study 3: Altering the Dropout Layer and Dense Layer

The performance of a classifier might vary depending on the number of dense and dropout layers. In this study, we have experimented by adding or removing some dropout and dense layers. In configuration 1, our base model attained the highest accuracy of 82.42%, but also had the highest number of parameters (317,190) and the longest training time per epoch (21 s). After adding dense and dropout layers (configuration 2), our base model performed with the second-highest accuracy with 300,678 parameters and 20 s per epoch. For configuration 3, our model acquired 82.16% test accuracy with the lowest number of parameters, 284,166, and a training time of 19 s in each epoch. Though the third configuration acquired close to the highest accuracy, its training time and parameters were less than for the other configurations. Therefore, configuration 3 was chosen for further experiments.

Study 4: Altering the Dropout Layer and Dense Layer

A classification model’s effectiveness is affected by choice of activation functions. The performance of a model can be improved by choosing the best activation function. We experimented with several activation functions, including the hyperbolic tangent activation function (Tanh), the rectified linear unit (ReLU), the exponential linear unit (ELU), the soft sign, and the soft plus. Table 5 shows that the ReLU activation function performed very well, with the highest test accuracy of 83.06%. The number of parameters (284,166) and the training time for each epoch (19 s) are the same for all the activation functions. In this regard, the ReLU activation is chosen for further studies.

Study 5: Altering the type of pooling layer

A total of two different pooling layers, maxpooling and average pooling, were tried. In both configurations, the number of trainable parameters and the training time for each epoch were 284,166 and 47 s, respectively. The base model achieved the highest accuracy of 84.33% using the maxpooling layer. For this reason, the maxpooling layer was selected for further processing.

Study 6: Altering the stride size

This study investigates alternative stride sizes in the model’s transformer layers. We have applied several stride sizes: 1, 2, 3, and 4. In each configuration, the number of parameters was the same, 284,166, and each configuration took 19 s per epoch. The first configuration had the highest accuracy (84.33%), which was the same as in the previous study, while the other configurations resulted in lower accuracies. Stride size 1 was therefore selected for further processing.

Study 7: Altering the kernel size

Several transformer layer kernel sizes (4, 3, 2, and 1) were explored. A kernel size of 3 resulted in the highest test accuracy of 84.62% with 225,478 parameters, and its training time per epoch was 16 s. The first configuration obtained a similar accuracy (84.33%), while the accuracy dropped for other configurations. Therefore, a kernel size of 3 for the transformer layers was used for further ablation studies.

Study 8: Altering the kernel size of the pooling layer

Experiments were conducted with different kernel sizes (5, 4, 3, 2, and 1) for the pooling layers. The number of parameters (225,478) and the time per epoch (16 s) were the same for all kernel sizes. A kernel size of 3 resulted in the highest test accuracy of 85.68%. Most other configurations acquired close to the highest accuracy, while in some cases, the accuracy decreased. Therefore, a kernel size of 3 for the model’s pooling layers would be used in further ablation studies.

Study 9: Altering the loss function

To optimize the model, experiments were conducted with five loss functions: binary cross-entropy, categorical cross-entropy, mean squared error, mean absolute error, and mean squared logarithmic error. The outcomes are displayed in Table 6. All the parameters and the epoch time are the same for all configurations. Categorical cross-entropy outperformed the other loss functions with the highest accuracy of 87.35%. Therefore, the categorical cross-entropy loss function was used for further studies.

Study 10: Altering the batch size

The classification performance may change with different batch sizes. We, therefore, experimented with several batch sizes: 256, 128, 64, and 32. The number of parameters remained the same (225,478). Though the model achieved the highest accuracy of 87.78% while using a batch size of 32, the per-epoch time was long at 31 s. For configuration 1, the model had close to the highest accuracy with minimal training time per epoch (12 s). We, therefore, chose a batch size of 256 for the remaining ablation studies.

Study 11: Altering the optimizer

A total of five different optimizers, Adam, Nadam, SGD, Adamax, and the RMSprop, were explored to identify the optimal optimizer. In this case, the Adam optimizer attained the highest accuracy, 87.42%, while the other optimizers obtained close to the highest accuracy. The Adam optimizer was selected for further experimentation.

Study 12: Altering the learning rate

We experimented with learning rates of 0.01, 0.006, 0.001, and 0.0008. The number of parameters and the training time were the same. The model achieved the highest accuracy while using a learning rate of 0.001. We, therefore, selected a learning rate of 0.001 for our proposed model.

The configuration of the proposed model after completing the ablation studies is shown in Table 7.

Figure 9 displays the configuration of our proposed model architecture. Figure 9 depicts the increase in test accuracy and the decrease in the number of parameters. Initially, the parameters were 541,638 with a test accuracy of 82.52%. After completing the ablation study, the accuracy increased by 7.65% (90.17%), and the number of parameters of the model decreased by 316,160.

### 3.7. DR-CCTNet

After performing ablation studies on the base CTT model, a new CTT model, named DR-CCTNet, was proposed. DR-CCTNet architecture is constructed to minimize computational complexity and training time while optimizing performance. The DR-CCTNet model resembles the original CCT model with fewer transformer encoder components and some layer changes. Figure 10 depicts the DR-CCTNet model architecture. The base or the standard CTT model has two transformer encoder blocks, whereas the DR-CCTNet model has only one transformer encoder block, resulting in a smaller model with a faster training time. Otherwise, the model architecture is similar but with several changes in model hyperparameters, such as the stride and kernel sizes. After ablation, the kernel and the stride size were replaced with 3 and 1, respectively. In the base CTT model, there were two pairs of dense and dropout layers, while in the DR-CCTNet, one pair of dense and dropout layers was eliminated. Moreover, the CCT tokenizer output was replaced with a 16 × 128 dimension. In contrast to transfer-based models, the model does not need positional encoding. This contributes to its low computational complexity. Self-attention has an O(m^2^.d) computational complexity, where m denotes the length of the input sequence, and d represents the dimensionality of the vector representation. Adding positional encoding raises the computational complexity to (O(m^2^.d + m.d^2^)) [35]. Since positional encoding is not required in the DR-CCTNet model and the transformer backbone is completely based on the self-attention system, the training and testing phases of the proposed model require fewer resources and are faster. This improves the model’s efficiency.

## 4. Analysis of Results

### 4.1. Performance Metrics

Several metrics are produced to evaluate the effectiveness of the proposed classification model. When the model correctly categorizes the positive class, the outcome is a true positive (*TP*). When the model correctly identifies the negative class, it is called the true negative (*TN*). A false positive (*FP*) is an outcome where the model incorrectly predicts the positive class, while a false negative (*FN*) is an outcome where the model incorrectly predicts the negative class. For this research, we calculated the accuracy (*ACC*), precision, recall, specificity, F1-score, false positive rate (*FPR*), false negative rate (*FNR*), false discovery rate (*FDR*), negative predicted value (*NPV*), and the Matthew correlation coefficient (*MCC*) to evaluate the model’s performance [27,34].
(6)$$ACC=\frac{TP+TN}{TP+TN+FP+FN}\;$$
(7)$$recall=\frac{TP}{TP+FN}\;$$
(8)$$specificity=\frac{TN}{TN+FP}$$
(9)$$precision=\frac{TP}{TP+FP}$$
(10)$$F_{1}-score=2\frac{precision\times recall}{precision+recall}\;$$
(11)$$FPR=\frac{FP}{FP+TN}\;\;$$
(12)$$FNR=\frac{FN}{FN+TP}\;$$
(13)$$FDR=\frac{FP}{FP+TP}$$
(14)$$\;NPV=\frac{TN}{TN+FN}\;\;$$
(15)$$MCC=\frac{TP\times TN-FP\times FN}{\sqrt{\left(TP+FP\right)\left(TP+FN\right)\left(TN+FP\right)\left(TN+FN\right)}}\;$$

### 4.2. Performance Analysis of the Proposed Model

After performing ablation studies on the base model, a final proposed CCT model with a significantly improved classification performance was obtained. Table 8 shows the values of several performance metrics, including statistical analysis, for the proposed CCT model.

Table 8 shows that after evaluating the proposed CTT model with the test data set, the model obtained a precision of 89.38%, a recall of 90.10%, a specificity of 97.51%, and an F1-score of 89.72%. The FPR, FNR, FDR, NPV, and MCC were 0.02492, 0.09526, 0.10618, 97.46%, and 87.22%, respectively. The values of the performance metrics indicate that our proposed model can effectively classify fundus images.

Figure 11 shows the accuracy and loss curves of the proposed model. The training and validation curves converge efficiently with no significant gaps between them, indicating no overfitting during the model’s training phase (Figure 11A). Similarly, the loss curves in Figure 11B demonstrate consistent convergence. It may be inferred that no overfitting or underfitting occurred throughout the model’s training phase. Figure 11C depicts the proposed model’s confusion matrix. The row values denote the actual labels of the test images, while the column values indicate the labels predicted by the model. The diagonal values of the confusion matrix show the number of correctly predicted test images. It can be seen that the model is not biased toward one or more classes and does not predict any one class much better than the others. In fact, the model provides almost a similar number of correct predictions for each class.

### 4.3. Analysis of Image Size Tuning

Table 9 shows the results of the proposed model for images of different sizes. For comparison accuracy, the number of parameters and the execution time were also considered. The results listed in Table 5 and Table 6 demonstrate that we can obtain optimal results using an image size of 16 × 16. To verify our ablation findings and display the computation times, we changed the image size while keeping the parameters and configuration the same as listed in Table 7. We first changed the image size to 32 × 32 and acquired a higher accuracy of 93.81%. However, this image size was the most time-consuming, which does not align with this study’s objective to minimize computational training time. The size was then changed to 28 × 28 and 24 × 24. Compared to the 32 × 32 images, the accuracy was decreased. The accuracy for the 28 × 28 dimension was 93.33%, which is higher than the accuracy for our 16 × 16 images, but this required 12 s more per epoch. The model yielded an accuracy of 91.12% for 24 × 24 images, which is only 0.54% more than for 16 × 16 images, but this also cost nearly 7 s more time for each epoch. The experiment demonstrates that the image size somewhat increases the overall accuracy but demands more computational time. This cost will increase for a large amount of data. In our study, we aimed to develop an efficient architecture for diagnosing retinopathy images. Since the other three dimensions take a considerably longer execution time, we preferred to utilize the 16 × 16 image size for the configuration.

### 4.4. Analysis of the Performance with Image Reduction

This study has followed an image reduction strategy to test the robustness of the proposed DR-CCTNet model’s performance. The number of input images gradually decreased, and the DR-CCTNet model was tested multiple times to observe the accuracy and an error bar to visualize the gaps between the actual and estimated results. The number of images was reduced by 25% of the original size for each step. Additionally, we performed three times on randomly selected images to test the model’s performance in each test case. Figure 12 depicts the error bars with the mean value of model accuracy three times the test case. In contrast, the ± value indicates the difference between the maximum accuracy (in triplicate performance) and their mean value. The results are depicted in Figure 12.

Figure 12 demonstrates that during the test cases, the model obtained a ratio of 90.13 ± 0.04 % for 100% of images, 88.667 ± 0.155 % for 75% of images, 86.19 ± 0.232 % for 50% of images, and 84.79 ± 0.207 % for 25% images. Figure 12 shows that when the data is reduced to 75% and 50%, the accuracy does not drop significantly compared to the highest accuracy of 90.17%. In the final experiment, we only used 25% of the data compared to the original augmented dataset, yet the drop in accuracy is only 5.377% (compared to the 90.17% accuracy); it might be increased or decreased to 0.207. The finding of this experiment illustrates that even when 75% of the images are removed, our model still performs well in terms of accuracy. It demonstrates the robustness of our proposed DR-CCTNet model.

## 5. Conclusions

This study addresses some issues of automated DR diagnosis. A modified transformer model (DR-CCTNet) framework was proposed. A wide range of fundus images was considered; five different datasets with different resolutions and quality of images were combined. Since image qualities were diverse and various artifacts and noise were present in these images, further processing was challenging. Image processing techniques were adopted. Working with a large amount of data, 154,882 images, was another challenge of this study. The dataset was balanced and augmented. From three transformers and four transfer learning models, the best model, CCT, was selected. A total of 12 ablation studies were performed to identify the best configuration and improve the model’s classification accuracy; the DR-CCTNet model was proposed after modifying the original CCT model to address training time concerns and work with a large data hub. An image size of 16 × 16 resulted in the lowest computational time. Our proposed model achieved an accuracy of 90.17%, even with low-pixel images, and still displayed a strong performance with fewer images, indicating that the model was robust. This study’s contributions include a larger and more reliable dataset created with a unique augmentation strategy, quality of enhancement of the fundus images using different image pre-processing techniques, a detailed comparison between three transformer and four transfer learning models, and an ablation study to propose an optimized model that can produce an excellent result with 16 × 16 size images.

## 6. Limitations and Future Research

The proposed transformer (DR-CCTNet) model performed substantially better than conventional deep learning models for multiclass identification of a large variety of low-pixel fundus images. Nonetheless, our proposed model has drawbacks that can be addressed in future research. Increasing the number of raw images for all grades might be possible in the future. The performance of our proposed model on real-time data could also be assessed. However, our proposed method operates well in most test instances and can accurately classify the five distinct fundus image classes. The proposed DR-CCTNet model is robust despite minor shortcomings.

## Figures and Tables

**Figure 1 biomedicines-11-01566-f001:**
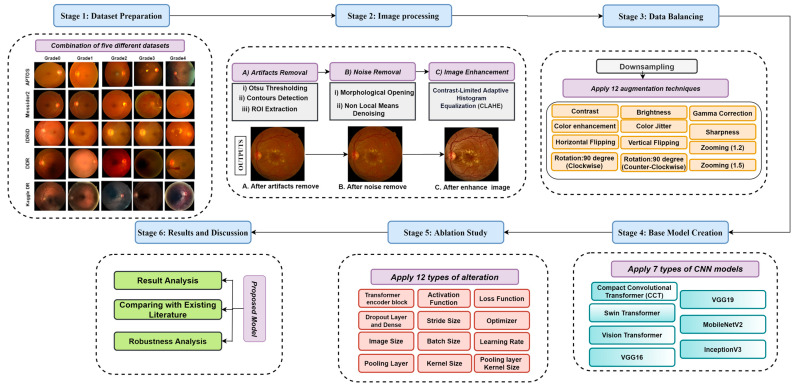
Workflow diagram; Stage-1 (Dataset Preparation): Combination of five different datasets; Stage-2: Image pre-processing (A). Artifacts removal, (B). Noise Removal, and (C). Image Enhancement; Stage-3 (Data Balancing): Down sampling (Applying 12 types of augmentation techniques); Stage-4 (Base Model Creation): Apply seven types of CNN models (Compact Convolutional Transformer (CCT), Swin Transformer, Vision Transformer, VGG16, VGG19, MobileNetV2, and InceptionV3); Stage-5 (Ablation Study): Apply twelve types of alteration to the base model for creating a proposed model; Stage-6 (Results and Discussion): Results Analysis, Comparison with Existing Literature, model performance evaluation, and Robustness Analysis.

**Figure 2 biomedicines-11-01566-f002:**
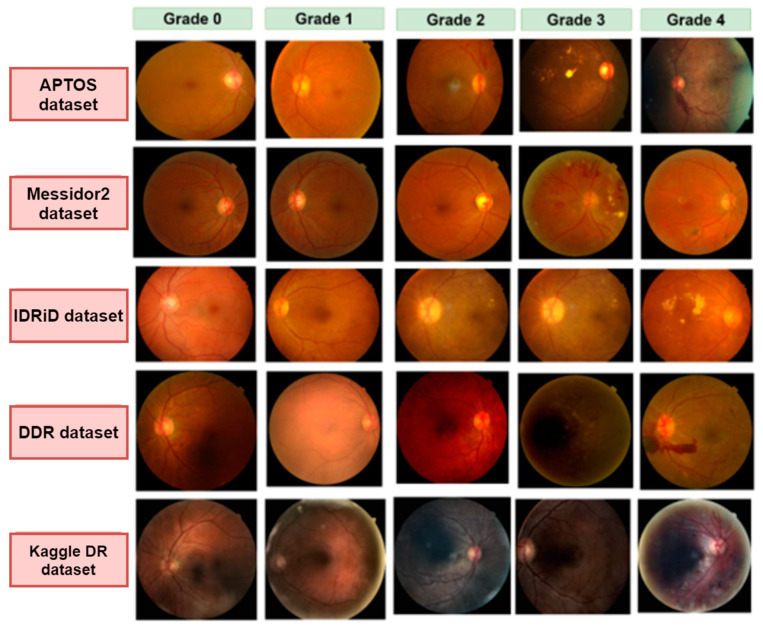
Examples of five grades (grade 0–grade 4) images for five different datasets: APTOS, Messidor2, IDRiD, DDR, and Kaggle DR.

**Figure 3 biomedicines-11-01566-f003:**
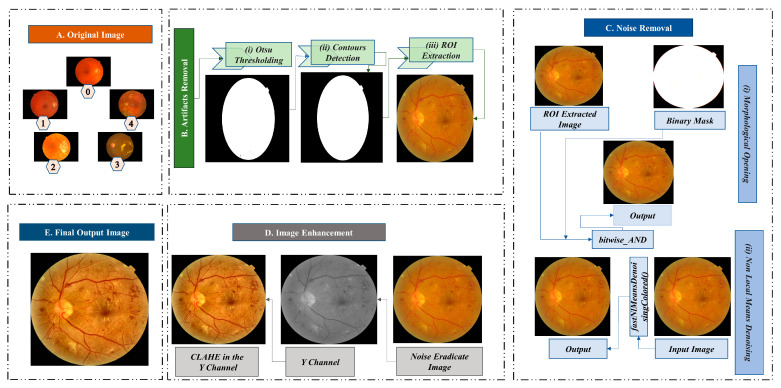
Image pre-processing techniques; Block (**A**): Original image (Grade 0: no DR, Grade 1: mild NPDR, Grade 2: moderate NPDR, Grade 3: severe NPDR, and Grade 4: PDR), Block (**B**): Artifacts removal (i. Otsu Thresholding, ii. Contours Detection, and iii. ROI Extraction), Block (**C**): Noise removal, Block (**D**): Image enhancement (CLAHE in the Y channel), and Block (**E**): Final output image.

**Figure 4 biomedicines-11-01566-f004:**
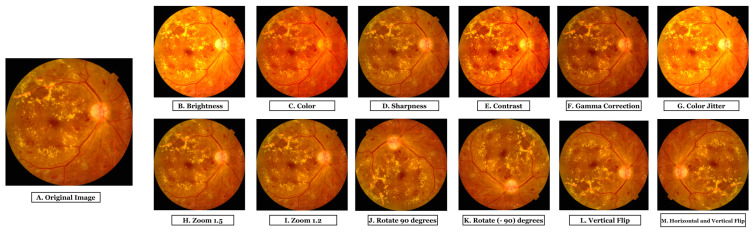
Output images after applying 12 augmentation techniques; (**A**). Original Image, (**B**). Brightness, (**C**). Color, (**D**). Sharpness, (**E**). Contrast, (**F**). Gamma Correction, (**G**). Color Jitter, (**H**). Zoom (1.5), (**I**). Zoom (1.2), (**J**). Rotate 90 degrees, (**K**). Rotate (−90) degrees, (**L**). Vertical Flip, and (**M**). Horizontal Vertical Flip.

**Figure 5 biomedicines-11-01566-f005:**
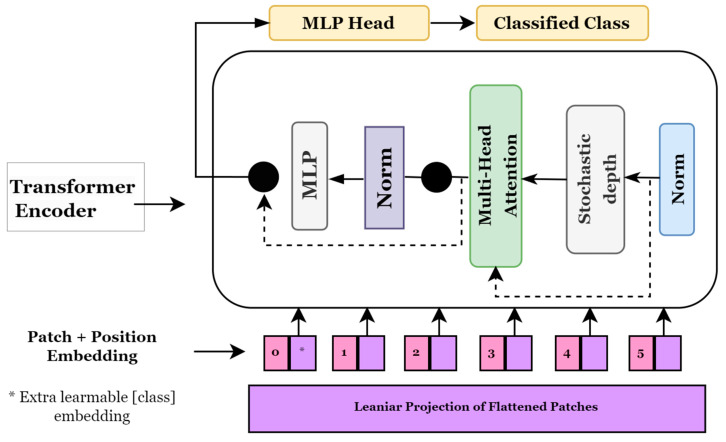
Vision Transformer model architecture.

**Figure 6 biomedicines-11-01566-f006:**
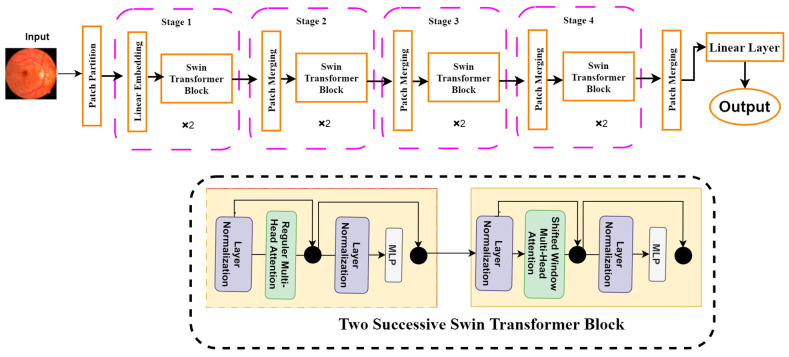
Swin Transformer model architecture.

**Figure 7 biomedicines-11-01566-f007:**
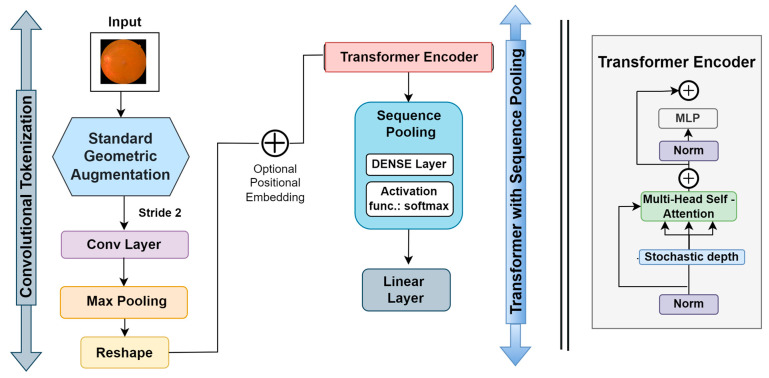
CCT model architecture.

**Figure 8 biomedicines-11-01566-f008:**
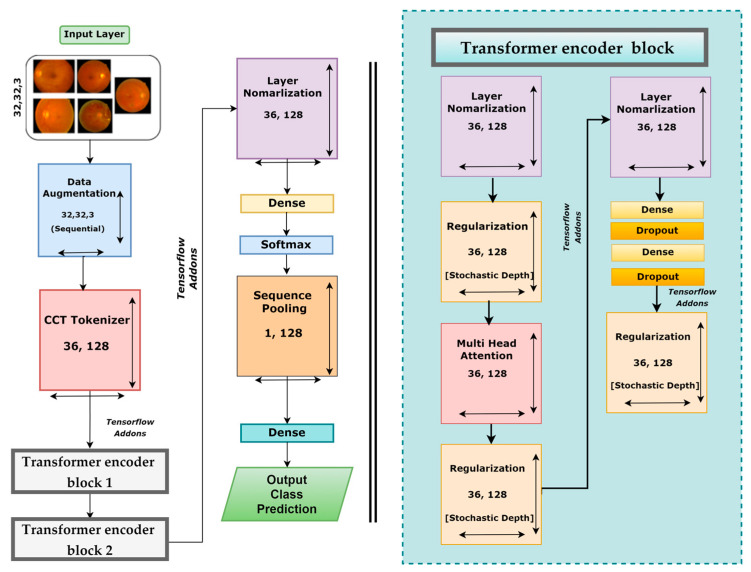
Base model architecture of CCT.

**Figure 9 biomedicines-11-01566-f009:**
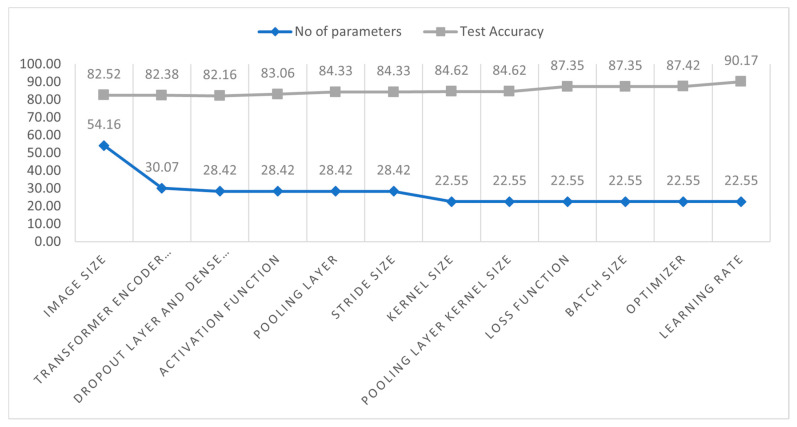
The number of parameters (scaled from 0 to 100) and test accuracy for each study.

**Figure 10 biomedicines-11-01566-f010:**
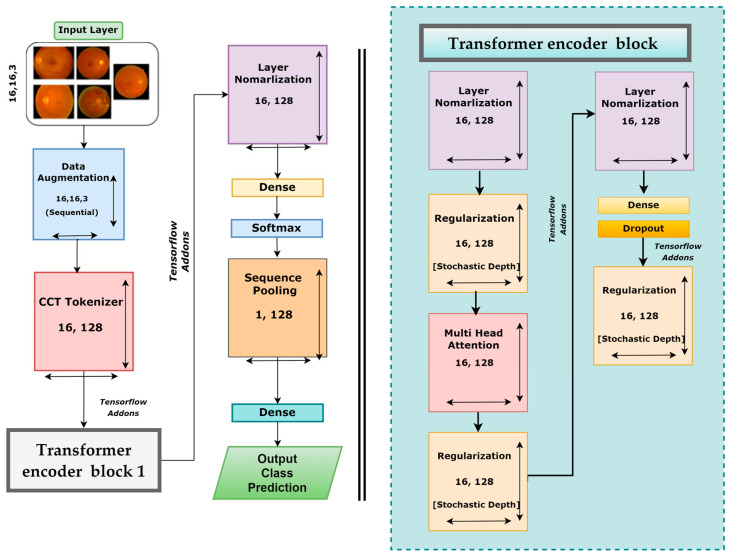
Architecture of the proposed DR-CCTNet model.

**Figure 11 biomedicines-11-01566-f011:**
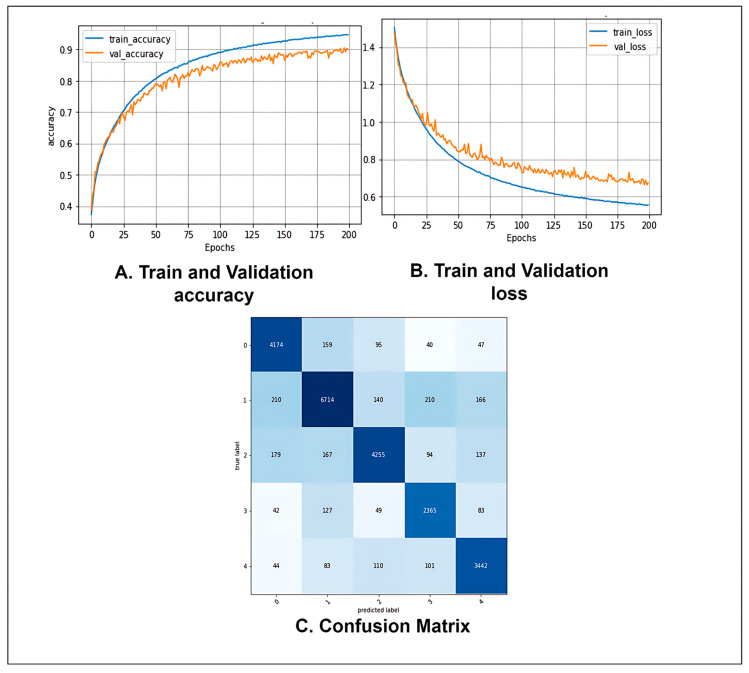
(**A**) Accuracy curve, (**B**) Loss curve, and (**C**) Confusion matrix of the proposed CCT model after ablation studies.

**Figure 12 biomedicines-11-01566-f012:**
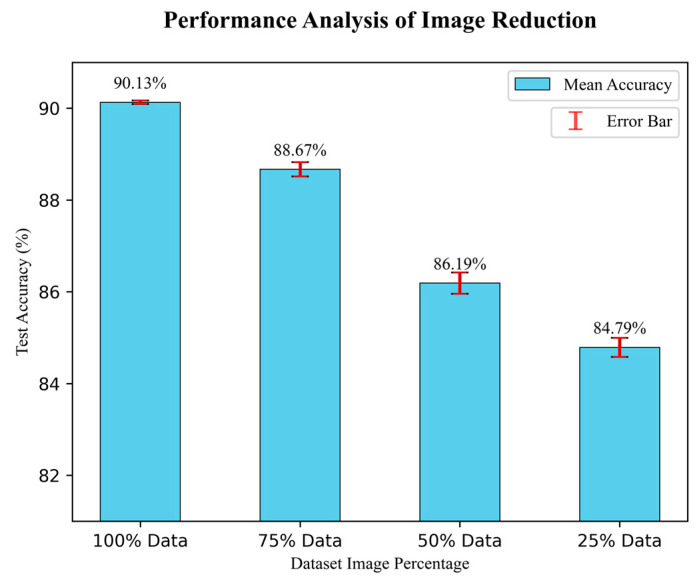
Results of image reduction.

**Table 1 biomedicines-11-01566-t001:** Comparison of existing literature.

Paper	Model	Datasets	Number of Image	Image Size (Pixels)
Shanthi et al. [3]	Modified AlexNet	Messidor	1190	259 × 259
Hemanth et al. [10]	Proposed CNN model	MESSIDOR	400	150× 255
Gu et al. [11]	Proposed transformer model by using vision transformers and residual attention	DDR and IDRiD	13,673 and 516	512 × 512
Sehrish et al. [12]	Ensemble Classifier	DR’s Kaggle dataset	35,126	512 × 512
Liu et al. [13]	Hybrid deep learning model	DeepDR, APOTS, and EyePACS	1200, 3662 and 35,126	380 × 380
Wu et al. [14]	CF-DRNet	IDRiD and DR’s Kaggle dataset	413 and 35,126	256 × 256
Lam et al. [15]	CNN based architectures	DR’s Kaggle dataset and Messidor-1	35,126 and 1090	256 × 256
Gao et al. [16]	Inception@4	DR Fundus	4476	600 × 600

**Table 2 biomedicines-11-01566-t002:** Dataset description.

Datasets	Messidor2	APTOS	IDRiD	Diabetic Retinopathy	DDR	Merge
No DR (0)	1017	1805	134	25,810	6265	34,830
Mild NPDR (1)	270	370	20	2443	630	3718
Moderate NPDR (2)	347	999	136	5292	4447	11,209
Severe NPDR (3)	75	193	74	873	236	1441
PDR (4)	35	295	49	708	913	1987
Total Images	1744	3662	413	35,126	12,491	53,185

**Table 3 biomedicines-11-01566-t003:** Image counts after augmentation.

Grade	Merge	Under Sampling	After Augmentation
0	34,830	2384	30,992
1	3718	3718	48,334
2	11,209	2384	30,992
3	1441	1441	18,733
4	1987	1987	25,831
Total	53,185	11,914	154,882

**Table 4 biomedicines-11-01566-t004:** Result of augmentation and image size based on transformer and transfer learning models.

Model Name	Image Size	Accuracy	Epoch × Time
MobileNetV2	32 × 32	71.98%	195 × 386 s
VGG16	32 × 32	73.22%	165 × 391 s
VGG19	32 × 32	72.88%	177 × 404 s
ResNet50	32 × 32	76.67%	142 × 423 s
Swin Transformer	32 × 32	82.23%	200 × 95 s
Vit	32 × 32	81.56%	200 ×104 s
CCT	32 × 32	84.52%	200 × 78 s

**Table 5 biomedicines-11-01566-t005:** Ablation study on image size changing transformer encoder block dense layer, dropout layer, activation function, pooling layer, and stride size.

Study 1: Changing Image Size					
Configuration No.	Image Size	No. of Parameters	Epoch	Training Time	Test Accuracy (%)
1	32 × 32 (Base model)	5,41,638	200	78 s	84.88%
2	28 × 28	541,638	200	60 s	83.91%
3	24 × 24	541,638	200	48 s	83.23%
4	16 × 16	541,638	200	38 s	82.52%
Study 2: Changing the Transformer Encoder Block					
Configuration No.	No of transformer encoder blocks	No. of Parameters	Epoch	training time	Test accuracy (%)
1	3	707,142	200	70 s	82.88
2	2	541,638	200	38 s	82.52
3	1	300,678	200	21 s	82.38
Study 3: Changing the Dropout Layer and Dense Layer					
Configuration No.	No of dropout layer	No of dense layer	No. of Parameters	Epoch × training time	Test accuracy (%)
1	3	3	317,190	200 × 21 s	82.42
2	2	2	300,678	200 × 20 s	82.38
3	1	1	284,166	200 × 19 s	82.16
Study 4: Changing the activation function					
Configuration No.	Activation function	No. of parameters	Epoch × training time	Test accuracy (%)	Findings
1	Tanh	284,166	200 × 19 s	82.16	Previous accuracy
**2**	relu	284,166	200 × 19 s	83.06	Highest Accuracy
3	elu	284,166	200 × 19 s	82.38	Accuracy improved
4	softsign	284,166	200 × 19 s	76.40	Accuracy dropped
5	softplus	284,166	200 × 19 s	75.97	Accuracy dropped
Study 5: Changing the pooling layer					
Configuration No.	Type of pooling layer	No of parameters	Epoch × training time	Test accuracy (%)	Findings
**1**	Max	284,166	200 × 19 s	84.33	Highest Accuracy
2	Average	284,166	200 × 19 s	83.06	Previous accuracy
Study 6: Changing the stride size					
Configuration No.	No. of strides	No. of Parameters	Epoch × training time	Test accuracy (%)	Findings
**1**	**1**	284,166	200 × 19 s	84.33	Previous Accuracy
2	2	284,166	200 × 19 s	83.16	Accuracy dropped
3	3	284,166	200 × 19 s	82.97	Accuracy dropped
4	4	284,166	200 × 19 s	81.43	Accuracy dropped

**Table 6 biomedicines-11-01566-t006:** Ablation study on changing kernel size, kernel size of the pooling layer, loss function, and batch size.

Study 7: Changing the Kernel Size					
Configuration No.	No. of Kernel Size	No. of Parameter	Epoch × Training Time	Test Accuracy (%)	Finding
1	4	284,166	200 × 19 s	84.33	Previous accuracy
2	3	225,478	200 × 16 s	84.62	Highest Accuracy
3	2	183,558	200 × 13 s	80.12	Accuracy dropped
4	1	158,406	200 × 11 s	76.31	Accuracy dropped
Study 8: Changing the kernel size of the pooling layer					
Configuration No.	No. of pooling kernel size	No. of Parameter	Epoch × training time	Test accuracy (%)	Finding
1	5	225,478	200 × 16 s	84.57	Accuracy dropped
2	4	225,478	200 × 16 s	85.12	Accuracy improved
3	3	225,478	200 × 16 s	85.68	Highest Accuracy
4	2	225,478	200 × 16 s	84.62	Previous accuracy
5	1	225,478	200 × 16 s	83.9	Accuracy dropped
Study 9: Changing the loss function					
Configuration No.	Loss Function	No. of Parameter	Epoch × training time	Test accuracy (%)	Finding
1	Binary Cross-entropy	225,478	200 × 16 s	87.12	Accuracy improved
2	Categorical Cross-entropy	225,478	200 × 16 s	87.35	Highest Accuracy
3	Mean Squared Error	225,478	200 × 16 s	85.68	Previous accuracy
4	Mean absolute error	225,478	200 × 16 s	84.93	Accuracy dropped
5	Mean squared logarithmic error	225,478	200 × 16 s	85.76	Accuracy dropped
Study 10: Changing the batch size					
Configuration No.	Batch size	No. of Parameter	Epoch × training time	Test accuracy (%)	Finding
1	256	225,478	200 × 12 s	86.88	Accuracy dropped
2	128	225,478	200 × 16 s	87.35	Previous accuracy
3	64	225,478	200 × 22 s	87.53	Accuracy improved
4	32	225,478	200 × 31 s	87.78	Accuracy improved
Study 11: Changing the optimizer					
Configuration No.	Optimizer	No. of Parameter	Epoch × training time	Test accuracy (%)	Finding
1	Adam	225,478	200 × 16 s	87.42	Highest Accuracy
2	Nadam	225,478	200 × 16 s	86.78	Accuracy dropped
3	SGD	225,478	200 × 16 s	87.35	Previous accuracy
4	Adamax	225,478	200 × 16 s	84.18	Accuracy dropped
5	RMSprop	225,478	200 × 16 s	86.8	Accuracy dropped
Study 12: Changing the learning rate					
Configuration No.	Learning rate	No. of Parameter	Epoch × training time	Test accuracy (%)	Finding
1	0.01	225,478	200 × 16 s	86.12	Accuracy dropped
2	0.006	225,478	200 × 16 s	87.42	Previous accuracy
3	0.001	225,478	200 × 16 s	90.17	Highest Accuracy
4	0.0008	225,478	200 × 16 s	89.8	Accuracy improved

**Table 7 biomedicines-11-01566-t007:** Configuration of proposed architecture after the ablation study.

Configuration	Value
Image size	16 × 16
Epochs	100
Optimization function	Adam
Learning rate	0.001
Batch size	256
Kernel size	3
Activation function	ReLU
Loss Function	Categorical Cross-Entropy
Kernel size of the pooling layer	3
Stride size	1
Pooling layer	Max pooling
Projection_dim	128
Stochastic_depth_rate	0.1
Weight_decay	0.001

**Table 8 biomedicines-11-01566-t008:** Various matrices computed for performance evaluation of the proposed model.

Measure	Value
Recall	90.10%
Specificity	97.51%
Precision	89.38%
F1 Score (F1)	89.72%
Fall-out or False Positive Rate *(FPR)*	0.02492
Miss Rate or False Negative Rate *(FNR)*	0.09526
False Discovery Rate *(FDR)*	0.10618
Negative Predictive Value *(NPV)*	97.46%
Matthews Correlation Coefficient *(MCC)*	87.22%

**Table 9 biomedicines-11-01566-t009:** Robustness validation with different image sizes.

No. of Parameters	Image Size	Accuracy	Epoch × Time
225,478	32 × 32	93.81%	200 × 32 s
225,478	28 × 28	93.33%	200 × 28 s
225,478	24 × 24	91.12%	200 × 23 s
225,478	16 × 16	90.17%	200 × 16 s

## Data Availability

We employed five datasets, including APTOS [16], Messidor2 [17], IDRiD [18], DDR [19], and Kaggle Diabetic Retinopathy [20]. These datasets are publicly available.

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
