# Peer review of "A Computer-Aided Diagnostic System to Identify Diabetic Retinopathy, Utilizing a Modified Compact Convolutional Transformer and Low-Resolution Images to Reduce Computation Time"

_biomedicines, 2023, doi:10.3390/biomedicines11061566_

Round 1

Reviewer 1 Report

1- There are eight keywords that seem too many. reduce the number of keywords to no more than 6.

2- NPDR stands for what? write the full form of the abbreviation first. apply for whole the abbreviations of the manuscript.

3- Table 1. What is the unit for the image size?

4- Figure 1. Enlarge the size of stage 1, it is not visible in the current form.

5- The authors can use the following reference to improve some parts of the introduction: https://doi.org/10.1016/j.ijbiomac.2022.01.134

6- Figure 9. What is the difference between "Pooling Layer" and "Pooling Layer kernel size"?

7- Figure 12 needs error bars. try to repeat the test three times.

Minor editing of English language required.

Reviewer 2 Report

The authors of the article titled "A Computer-Aided Diagnostic System to Identify Diabetic Retinopathy, utilizing a Modified Compact Convolutional Transformer and Low-Resolution Images to Reduce Computation Time" present and analyze in a very detailed and scientifically sound manner the results of their research on identifying diabetic retinopathy with the help of artificial intelligence. The research has a high degree of originality, the introduction provides all the necessary background, the methods are described in a detailed and logical manner and the results are analyzed in relation to the other published data. The results of this research have a high potential to positively and significantly impact the diagnosis of diabetic retinopathy and it is of interest for computer science specialists and ophthalmologists as well. Given all of the above, the article is well suited for being published in "Biomedicines".

Round 2

Reviewer 1 Report

The manuscript is improved after answering the comments. It can be accepted in its current form.